# Prevalence and Risk Factors of Metabolic-Associated Fatty Liver Disease among 73,566 Individuals in Beijing, China

**DOI:** 10.3390/ijerph19042096

**Published:** 2022-02-13

**Authors:** Qianli Yuan, Huai Wang, Pei Gao, Weixin Chen, Min Lv, Shuang Bai, Jiang Wu

**Affiliations:** Institute for Immunization and Prevention, Beijing Center for Disease Prevention and Control, Beijing Research Center for Preventive Medicine, Beijing 100013, China; qianli240321@163.com (Q.Y.); 13811814942@163.com (H.W.); bjcdcgaopei@126.com (P.G.); chenweixin1985@sina.com (W.C.); 8872lm@163.com (M.L.); summer012388@hotmail.com (S.B.)

**Keywords:** metabolic-associated fatty liver disease, prevalence, risk factors, transient elastography

## Abstract

The prevalence of metabolic-associated fatty liver disease (MAFLD) is rarely reported in Beijing. The goal of this study was to estimate the prevalence and risk factors of MAFLD among Beijing adults aged ≥25 years old. A cross-sectional, community-based survey with multistage stratified cluster sampling was used. Demographic, transient elastography (TE), biochemical and blood examination information was collected in all the subjects in this study. The prevalence of MAFLD was 32.40% (23,832/73,566). Risk factors independently associated with MAFLD included male gender (OR = 1.47, 95%CI, 1.43–1.52), urban residence (OR = 1.06, 95% CI, 1.02–1.10), older age (30–39 years: OR = 1.29; 40–49 years: OR = 1.43; 50–59 years: OR = 1.09; ≥60 years: OR = 1.52) and lower education (middle school: OR = 2.03; high school: OR = 1.89; undergraduate: OR = 1.69). MAFLD was more common in females than in males after 50 years of age. Lean/normal weight MAFLD patients account for approximately 3.04% (724/23,832) of MAFLD. Compared to non-MAFLD subjects, the lean/normal MAFLD patients had a higher prevalence of hypertension and diabetes, and had a higher degree of hepatic steatosis and liver function enzymology parameters (all *p* < 0.001). MAFLD was highly prevalent among the general population aged ≥25 years old in Beijing. MAFLD was closely associated with male gender, older age, lower education and urban residence. Even lean/normal-weight people were under risk of MAFLD.

## 1. Introduction

Metabolic-associated fatty liver disease (MAFLD), a new term changed from non-alcoholic fatty liver disease (NAFLD) in 2020 [1], is one of the most important causes of chronic liver diseases. MAFLD is a spectrum ranging from simple steatosis to nonalcoholic steatohepatitis, which can progress to cirrhosis [2,3]. Obesity, Type 2 diabetes mellitus (DM), and metabolic syndrome are identified as the most important risk factors for MAFLD.

Over the past two decades, MAFLD has become the most common chronic liver disease and the global prevalence of MAFLD is 25.24%, with the highest prevalence in the Middle East (31.79%) and South America (30.45%), followed by Asia (27.37%), North America (24.13%), and Europe (23.71%), whereas MAFLD is less common in Africa (13.48%) [4,5,6]. In China, the increase in the prevalence of MAFLD was from 23.8% in 2001 to 32.9% in 2018 [7], and it is continuing to increase along with the increasing prevalence of metabolic syndrome, DM, cardiovascular disease and other chronic metabolic diseases. It is predicted that by 2030, the number of MAFLD cases will be around 314.58 million, suggesting the tremendous impacts of MAFLD in upcoming decades [8]. 

Liver biopsy (LB) remains the gold standard for the evaluation of hepatic fibrosis in patients with MAFLD. Nevertheless, LB is an invasive test with risk of complications and sampling error. Transient elastography (TE) was a promising technique for the clinical diagnosis and evaluation of hepatic steatosis and fibrosis [6]. 

It is very important to know the prevalence of MAFLD, since most patients with MAFLD have no obvious clinical symptoms, and long-term intrahepatic fat accumulation can promote the occurrence of liver fibrosis, until cirrhosis and even liver failure. Until now rare data was reported about the MAFLD prevalence in Beijing, so the objective of this study was to estimate the prevalence and risk factors of MAFLD in the adult population of Beijing.

## 2. Methods

### 2.1. Study Population and Data Collection

A cross-sectional, community-based survey was conducted from 28 July 2017 to 31 October 2019 in Beijing. It was based on a multistage stratified cluster sampling method. First, 11 out of 16 districts were randomly sampled. Second, 2 townships in each district were sampled with probability proportional to size (PPS) method. Third, 3–5 residents’ committees or villages in each township were sampled with PPS method. All residents within sampled residents’ committees or villages who were 25 years of age or older, possessed medical insurance and had lived in Beijing for over 6 months were asked to participate in the investigation. Finally, a total of 74,998 participants were involved in this study. All subjects were asked to sign an informed consent agreement after deciding to participate. A standardized, structured questionnaire was administered by trained investigators to collect sociodemographic information, alcohol consumption, medical history and factors related to MAFLD. After investigating, all subjects underwent routine physical examination, blood biochemical examination and liver examination. Systolic blood pressure (SBP) and diastolic blood pressure (DBP) were measured using an automated sphygmomanometer with the subject in a sitting position. The physical examination included physical test and FibroTouch test. 

### 2.2. Blood Collection and Testing

For collection, 5 ml fasting venous blood was extracted by trained health personnel from each participant after 12 h of fasting, and were tested on Hitachi 7600–110 automatic analyzer (Hitachi High-Technologies, Tokyo, Japan). Complete blood count was analyzed by Cell-DYN Ruby (Abbott Laboratories, Diagnostic Division, Abbott Park, IL, USA) within 2 h. Blood biochemical items including alanine aminotransferase, aspartate aminotransferase, fasting plasma glucose, gamma-glutamyltransferase, album, triglyceride, cholesterol, high-density lipoprotein, low-density lipoprotein cholesterol, hemoglobin A1c, alpha-fetoprotein, white blood cell, viral markers, etc., were examined. All normal value ranges established in accordance with the reagent instructions.

### 2.3. Transient Elastography

TE was performed using FibroTouch FT100 provided by Wuxi Hisky Medical Technologies, Wuxi, China. Liver steatosis and stiffness was performed according to the instructions provided by the manufacturer. Steatosis is quantitatively assessed by measuring the extent of attenuation of ultrasound signal occurs in liver, referred as the ultrasound attenuation parameter (UAP). Meanwhile, fibrosis is quantitatively assessed by measuring the speed of the shear wave propagation in liver, referred as liver stiffness measurement (LSM). The subject was at supine position with the right arm in maximal abduction to expand the intercostal space. An image-guided probe was used to detect the region through the seventh–ninth intercostal space. LSM and UAP were considered as reliable only if 10 successful measurements were obtained, and with an interquartile range/median of LSM and UAP of 30% and a success rate of 60%. Fat attenuation value ≥244 dB/m were considered as fatty liver [6].

### 2.4. Diagnostic Criteria

#### 2.4.1. MAFLD 

MAFLD was diagnosed based on FibroTouch examination and the presence of any one of the following three conditions, namely overweight/obesity, presence of DM or evidence of metabolic dysregulation [1]. The metabolic dysregulation was defined as the presence of two or more of the following conditions: (1) waist circumference ≥ 102 in men and 88 cm in women; (2) blood pressure ≥ 130/85 mmHg (SBP/DBP) or specific drug treatment; (3) TG ≥ 1.70 mmol/L or specific drug treatment; (4) HDL-C < 1.0 mmol/L for males and <1.3 mmol/L for females; (5) prediabetes (i.e., fasting glucose (FPG) levels 5.6 to 6.9 mmol/L, or 2 h post-load glucose levels 7.8 to 11.0 mmol/L or HbA1c 5.7% to 6.4%; (6) homeostasis model assessment insulin resistance (HOMA-IR) score ≥ 2.5; (7) C-reactive protein (CRP) level > 2 mg/L. The non-MAFLD population referred to patients who do not meet the above conditions.

#### 2.4.2. BMI

Body mass index (BMI) was calculated as height (m) divided by weight (kg) squared(m/kg^2^). We classified BMI according to the criteria proposed for Asian populations: lean (BMI < 23 kg/m^2^), overweight (BMI 23.0–24.9 kg/m^2^) or obese (BMI ≥ 25.0 kg/m^2^). Therefore, participants were divided into four groups: lean/normal weight MAFLD, overweight MAFLD, obese MAFLD, and non-MAFLD.

#### 2.4.3. Hypertension and DM

Subjects with their FPG value of ≥7.0 mmol/L and/or with a history of DM were considered to have DM. Hypertension was defined as SBP ≥ 140 mmHg or DBP ≥ 90 mmHg without history of taking antihypertensive medication in the past 6 months.

### 2.5. Statistical Analysis

Epidata 3.1 was used for data entry. Statistical analyses were performed using SPSS version 21.0 (SPSS, Chicago, IL, USA). Continuous variables were expressed as mean ± standard deviation and categorical variables were expressed as frequencies. The significance of the two independent samples was analyzed by the unpaired t test. Categorical variables were compared using Chi-square test (*χ*^2^) for trend. Multiple logistic regression analysis with probability for entry 0.05 and removal 0.1 was performed to identify factors associated with MAFLD. Variables including gender, age, residence, education and daily alcohol consumption were included in analysis, while variables involved in the MAFLD definitions such as SBP, DBP, FPG, TG, HDL-c and WC were not included in the multi-analysis. A forward LR variable selection method was used. Statistical significance is determined as *p* < 0.05. 

## 3. Results

### 3.1. Characteristics of Study Population

In this study were involved 74,998 participants, with a response rate of 98.40%. After excluding subjects with invalid FibroTouch measurements (1332), missing value of WC or weight (9) as well as data of FPG, SBP or DBP with logical errors (91), eventually 73,566 participants were included in the final analysis. The mean age of the subjects was 47.20 ± 14.11 years. A total of 39,726 (54.00%) of the subjects were female. A total of 9051 (12.30%) and 23,771 (32.3%) subjects had DM and hypertension respectively, and 53,699 (73.00%) subjects were overweight or obese (Table 1).

### 3.2. Age and Gender-Specific Prevalence of MAFLD

The overall prevalence of MAFLD was 32.40% (23,832/73,566). There was prevalent difference between males and females (36.80% vs. 28.65%, *p* < 0.001). The prevalence of MAFLD in male and female both increased with age (*p* < 0.001). The prevalence of MAFLD in male peaked at the 40–49 years, and then began to decline. MAFLD was more common in females than in males after 50 years of age (Figure 1).

### 3.3. Comparison of Blood Parameters among MAFLD and Non-MAFLD Participants

High levels of TG, FPG, CHOL, LDL-C, ALT, AST, TP, ALP, GGT, TBA, CHE, PA, RBC, HCG, HCT, MCH, MCHC, PLT, RDW, PLT, PDW, PCT and HbA1c were all positively associated with MAFLD (*p* < 0.001), but HDL-C, ALB, TBIL, DBIL and IBIL were negatively related with MAFLD (*p* < 0.001). MCHC, MPV and AFP did not reach significance (*p* > 0.05). Compared with subjects without MAFLD, the MAFLD patients drank more alcohol daily and had higher value of LSM and UAP (*p* > 0.05) (Table 2).

### 3.4. Factors Associated with Fatty Liver MAFLD

A multivariate regression logistic analysis was performed to identify the risk factors. The results showed that male gender, urban residence and older age were found to be independent risk factors for MAFLD, and higher education level was a protective factor for MAFLD (Table 3). Daily alcohol consumption was not significantly related to MAFLD (*p* > 0.05).

### 3.5. Comparison of the Blood Parameters between MAFLD Participants with Different BMI

In the general population, 3.72% had nonobese MAFLD. The lean/normal weight MAFLD group had a female predominance than the non-MAFLD group. With the increase in BMI of the MAFLD, the prevalence of hypertension and MD increased gradually. The prevalence of hypertension and DM in lean/normal weight MAFLD participants was higher than that in non-MAFLD participants. Compared with non-MAFLD participants, biochemical indicators including ALT, AST, TP, ALP, GGT, TBA, TG, CLU, FPG, CHE and PA were higher, and hematological parameters including HCB, HbA1c%, PCT and PLT were lower in lean/normal weight MAFLD group (all *p* < 0.001). Higher LSM and UAP value were observed in participants with lean MAFLD than in the non-MAFLD group (*p* < 0.001) (Table 4). 

## 4. Discussion

With the improvement of the economy, changes in living habits and diet structure, urbanization, changes in screening and diagnostic instruments and research methodology, the prevalence of MAFLD gradually increases. A meta-analysis published in 2014 indicated the prevalence of MAFLD in Chinese people older than 18 years is 20.09% (95%CI: 17.95–22.31%), and the pooled prevalence estimate risen over time [2]. A study showed the prevalence of MAFLD was 35.47% in 2006 and went up to 46.46% in 2014 in the same population in Zhejiang province after follow-up for 8 years [9]. This study showed that the MAFLD prevalence was 32.40% (23,832/73,566) in Beijing, which was similar to that in Henan (31.38%, 2017) [10] and higher than Shanghai (15.3%, 2005) [11], Guangdong (15%, 2007) [11], Chengdu (6.3%, 2009) [10], Jilin (15.5%, 2011) [11], Chongqing (26.1%, 2021) [12], Hong Kong (27.3%, 2012) [13] and Taiwan (11.5%, 2006) [14]. 

In this study, we found the prevalence of MAFLD increased with age in both males and females (*p* < 0.001), and MAFLD had a male predominance (36.80% vs. 28.65%). A possible reason is the difference in hormonal regulation between males and females. We also found the peak prevalence of MAFLD in men occurred earlier (40–49 years) than for women (over 60 years). This finding has also been reported in the literature [12,15]. Women enter their menopause after they are 50 years old. Estrogen is speculated to be able to suppress visceral fat accumulation and to increase subcutaneous fat accumulation. Previous studies have found that a decrease in estrogen in perimenopausal and postmenopausal women can lead to fat redistribution and thus cause metabolic disorders, including dyslipidaemia, glucose intolerance, and MAFLD [12,16]. The results indicated that estrogen might be a protective factor for females with MAFLD, and low estrogen levels during the postmenopausal periods may be an important risk factor for MAFLD in females.

Our data demonstrated that low level of education and urban residence were associated with a high risk rate of MAFLD. The prevalence of MAFLD was highest in those with middle school education. Education level affects the known rate of MAFLD knowledge of occurrence and prevention, which means people with high education may have less adverse factors such as eating imbalance and overweight. Urban residents have relatively higher MAFLD prevalence. It may be explained by their better life quality that were more likely to have unhealthy eating habits and overnutrition problem, which may lead to fat accumulation. Until now, the effect of alcohol consumption on MAFLD is unclear. Previous studies suggest that alcohol consumption is positively or negatively associated with MAFLD compared to abstinence [17,18,19,20], while our study found no association between alcohol consumption and MAFLD prevalence. The occurrence of this result did not exclude the possibility of confounding factors, because alcohol consumption was associated with some MAFLD risk factors, and those risk factors were not adjusted for in analyses.

Obesity is associated with MAFLD, however, NAFLD can also be observed in non-obese individuals and has different clinical characteristics. It was reported the prevalence varied from 15% to 21% in nonobese Asians [21]. MAFLD was considered as an early predictor of metabolic disorders and a major cause of cryptogenic liver disease in normal-weight populations. It is essential to distinguish normal weight and lean from overweight–obese MAFLD, so that specific treatment can be provided to halt or prevent the development of MAFLD. Till now, few studies focused on MAFLD in lean/normal weight individuals in China. In the current study, we classified the patients with MAFLD as lean/normal weight, overweight and obese MAFLD to define the prevalence and characterize the clinical biochemical, blood cell and metabolic features of MAFLD with different BMIs. In this study, the prevalence of MAFLD was 0.98% (724/73,566), 2.74% (2012/73,566) and 28.68% (21,096/73,566) in the lean/normal weight, overweight and obese groups, respectively. Although lean/normal weight MAFLD patients constituted a small proportion of MAFLD (3.04%, 724/23,832), they had more serious hepatic steatosis and stiffness than non-MAFLD patients. The occurrence and development of fatty liver usually were accompanied by the changes of various biochemical indicators. In this study, we found higher levels of liver function enzymology indexes and lipid metabolism parameters including higher ALT, AST, TP, DBIL, GGT, TBA, CHE, PA, TG, LDL-C and lower HDL-C in lean/normal weight MAFLD subjects, which indicated their higher risks in abnormality of hepatocellular function and dyslipidemia. ALT, AST and GGT are three kinds of liver function enzymology indexes that are commonly used in clinical practice. A large amount of fat deposition in liver cells can cause steatosis and pathological damage in liver cells, resulting in increased levels of ALT, AST, GGT and CHE. Elevated TG, decreased HDL-C, higher prevalence of hypertension and DM in the lean/normal weight MAFLD were all components of metabolic syndrome which have independent and important association with MAFLD [22]. Even people of lean/normal weight can be at risk for fatty liver, so BMI should not be the focus, the focus is how to prevent the occurrence of abnormal indicators. Having a proper diet and exercise actively were effective preventive measures for MAFLD. Furthermore, having regular physical examination, identifying and early intervening abnormal indicators related to liver function and lipid metabolism is not only conducive to prevent MAFLD, but also has important clinical value for the occurrence of cardiovascular and cerebrovascular diseases.

The first limitation of our study is the inability to determine the causal relationship between MAFLD and influencing factors due to limitations of cross-sectional surveys. Secondly, some important parameters that should have been involved in diagnosis of MAFLD, including 2 h post-load glucose levels, HOMA-IR score and CRP, were not available, which may lead to underestimated of the MAFLD prevalence. Thirdly, MAFLD was diagnosed using TE methods instead of histological assessments; nevertheless, TE methods are widely used for population-based studies. The strengths of this study are that this is the first study conducted in Beijing using the new definition of MAFLD, it was a community-based cross-sectional survey covering a large number of populations and the diagnosis of fatty liver was made by using TE; all these make the results convincing.

## 5. Conclusions

In conclusion, the prevalence of MAFLD among the general population aged ≥25 years old was 32.40% in Beijing. Male gender, old age, low education and urban residence were risk factors for MAFLD. Although lean/normal weight MAFLD constituted a small proportion of MAFLD, they had higher degree of hepatic steatosis and liver function enzymology indexes which was related to the occurrence and development of metabolic syndrome and fatty liver compared to the non-MAFLD subjects. People even with normal BMI should be aware of the risks of MAFLD.

## Figures and Tables

**Figure 1 ijerph-19-02096-f001:**
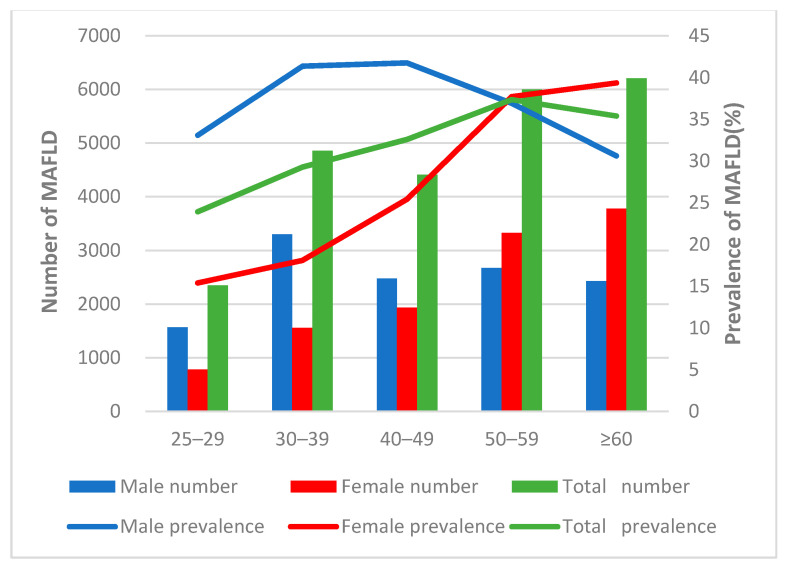
Prevalence of MAFLD by different age and gender.

**Table 1 ijerph-19-02096-t001:** Demographic characteristics of 73,566 participants.

Variable	Total (%)	Non-MAFLD (%)	MAFLD (%)	*p*
N	73,566	23,832	49,734	
Gender				
Male	33,840 (46.00)	21,388 (60.20)	12,452 (36.80)	<0.001
Female	39,726 (54.00)	28,346 (71.35)	11,380 (28.65)	
Age group				
25–29	9816 (13.34)	7467 (76.07)	2349 (23.93)	<0.001
30–39	16,589 (22.54)	11,731 (70.72)	4858 (29.28)	
40–49	13,545 (18.41)	9132 (67.42)	4413 (32.58)	
50–59	16,071 (21.85)	10,068 (62.65)	6003 (37.35)	
≥60	17,545 (23.85)	11,336 (64.61)	6209 (35.39)	
Residence				
Rural	32,381 (44.02)	28,014 (68.02)	13,171 (31.98)	0.007
City	41,185 (55.98)	21,720 (67.08)	10,661 (32.92)	
Education				
Middle school	30,796 (41.86)	19,798 (64.29)	10,998 (35.71)	<0.001
High school	17,913 (24.35)	11,944 (66.68)	5969 (33.32)	
Undergraduate	22,222 (30.21)	15,854 (71.34)	6368 (28.66)	
Graduate	2635 (3.58)	2138 (81.14)	497 (18.86)	
BMI				
Normal	19,867 (27.00)	19,143 (96.36)	724 (3.64)	<0.001
Overweight	14,722 (20.00)	12,710 (86.33)	2012 (13.67)	
Obese	38,977 (52.98)	17,881 (45.88)	21,096 (54.12)	
Hypertension				
No	49,795 (67.69)	13,437 (73.02)	36,358 (26.98)	<0.001
Yes	23,771 (32.3)	10,395 (56.27)	13,376 (43.73)	
DM				
No	64,515 (87.70)	19,128 (70.35)	45,387 (29.65)	<0.001
Yes	9051 (12.30)	4704 (48.03)	4347 (51.97)	

**Table 2 ijerph-19-02096-t002:** Comparison of blood parameters among MAFLD and non-MAFLD participants.

Blood Parameters	All	MAFLD	Non-MAFLD	*p*
Waist-to-hip ratio	0.87 ± 0.71	0.91 ± 0.06	0.86 ± 0.07	<0.001
ALT (IU/L)	22.35 ± 19.78	29.62 ± 25.92	18.87 ± 14.80	<0.001
AST(IU/L)	22.12 ± 12.99	24.58 ± 16.29	20.94 ± 10.86	<0.001
TP (g/L)	77.74 ± 4.90	78.17 ± 4.91	77.53 ± 4.88	<0.001
ALB (g/L)	48.70 ± 3.17	48.60 ± 3.28	48.75 ± 3.11	<0.001
TBIL (umol/L)	12.36 ± 5.17	12.20 ± 5.22	12.43 ± 5.14	<0.001
DBIL (umol/L)	3.62 ± 1.79	3.53 ± 1.83	3.66 ± 1.77	<0.001
IBIL (umol/L)	8.74 ± 3.78	8.68 ± 3.82	8.77 ± 3.75	0.001
ALP (IU/L)	86.08 ± 26.55	91.15 ± 26.56	83.65 ± 26.20	<0.001
GGT (IU/L)	31.61 ± 37.35	41.33 ± 42.32	26.95 ± 33.73	<0.001
TBA (umol/L)	3.82 ± 4.29	4.08 ± 4.64	3.70 ± 4.10	<0.001
CHE (IU/L)	8326.54 ± 1550.40	9021.44 ± 1472.02	7993.57 ± 1474.81	<0.001
PA (g/L)	0.29 ± 0.05	0.30 ± 0.05	0.28 ± 0.05	<0.001
TG (umol/L)	1.66 ± 1.69	2.26 ± 2.23	1.37 ± 1.26	<0.001
FPG (mmol/L))	5.96 ± 1.81	6.40 ± 2.11	5.75 ± 1.60	<0.001
CHOL (mmo/L)	5.07 ± 1.02	5.23 ± 1.06	4.99 ± 0.99	<0.001
HDL-C (mmo/L)	1.38 ± 0.33	1.26 ± 0.29	1.43 ± 0.34	<0.001
LDL-C (mmo/L)	2.96 ± 0.84	3.14 ± 0.85	2.87 ± 0.82	<0.001
HbA1c (%)	5.56 ± 0.98	5.81 ± 1.14	5.44 ± 0.88	<0.001
RBC× (10^12^/L)	4.71 ± 0.46	4.82 ± 0.46	4.66 ± 0.45	<0.001
HGB (g/L)	135.89 ± 15.30	139.11 ± 14.92	134.34 ± 15.25	0.018
HCT (%)	0.40 ± 0.04	0.41 ± 0.04	0.39 ± 0.04	0.012
MCV (fl)	84.71 ± 4.59	84.48 ± 4.27	84.82 ± 4.73	<0.001
MCH (pg)	28.91 ± 3.47	28.91 ± 4.58	28.90 ± 2.78	0.002
MCHC (g/L)	340.95 ± 35.77	341.94 ± 53.71	340.48 ± 22.57	0.68
RDW (%)	0.11 ± 0.01	0.11 ± 0.01	0.11 ± 0.01	0.001
PLT (×10^9^/L)	218.14 ± 55.25	219.93 ± 58.53	217.28 ± 53.59	<0.001
PDW (%)	0.20 ± 0.01	0.20 ± 0.01	0.20 ± 0.01	0.003
MPV (fl)	6.40 ± 1.12	6.40 ± 1.11	6.41 ± 1.13	0.41
PCT (%)	0.14 ± 0.05	0.14 ± 0.07	0.14 ± 0.04	<0.001
WBC (×10^9^/L)	6.35 ± 1.66	6.75 ± 1.70	6.15 ± 1.60	<0.001
LYM (×10^9^/L)	2.15 ± 0.66	2.29 ± 0.68	2.08 ± 0.64	<0.001
MONO (×10^9^/L)	0.40 ± 0.16	0.42 ± 0.16	0.39 ± 0.16	<0.001
NEU (×10^9^/L)	3.60 ± 1.25	3.83 ± 1.27	3.50 ± 1.23	<0.001
EOS (×10^9^/L)	0.14 ± 0.14	0.15 ± 0.16	0.13 ± 0.13	<0.001
BASO (×10^9^/L)	0.06 ± 0.05	0.06 ± 0.06	0.06 ± 0.05	<0.001
AFP (ng/mL)	3.50 ± 9.49	3.66 ± 16.13	3.43 ± 2.91	0.06
Alcohol_consumption (g)	6.62 ± 22.13	8.07 ± 24.83	5.93 ± 20.68	<0.001
LSM (Kpa)	6.53 ± 2.83	7.41 ± 3.17	6.10 ± 2.55	<0.001
UAP (dB/m)	230.14 ± 38.33	275.04 ± 25.46	208.62 ± 20.81	<0.001

ALT—alanine aminotransferase; AST—aspartate aminotransferase; TP—total proten; ALB—album; BIL—total bilirubin; DBIL—total bilirubin; IBIL—total bilirubin; ALP—alkaline phosphatase; GGT—gamma-glutamyltransferase; TBA—total bile acids; CHE—cholinesterase; PA—polymerase acidic protein; TG—triglyceride; FPG—fasting plasma glucose; CHOL—cholesterol; HDL-C—high-density lipoprotein cholesterol; LDL-C—low-density lipoprotein cholesterol; HbA1c—hemoglobin A1C; RBC—red blood cell count; HGB—hemoglobin; HCT—hematocrit; MCV—erythrocyte mean corpuscular volume; MCH—mean corpuscular hemoglobin; MCHC—mean erythrocyte hemoglobin concentration; RDW—width of red blood cell volume distribution; PLT—platelets; PDW—width of platelet volume distribution; MPV—mean platelet volume; PCT—procalcitonin; WBC—white blood cell; LYM—lymphocytes count; MONO—monocytes count; NEUT—neutrophils count; EOS—eosinophils count; BASO—basophils count; AFP—alpha-fetoprotein; LSM—liver fibrosis; UAP—ultrasound attenuation parameter.

**Table 3 ijerph-19-02096-t003:** Multivariable analysis of MAFLD influencing factors.

Variables	Coefficient	Standard Error	Odds Ratio (95%CI)	*p* Value
Gender				
Female			Ref.	
Male	0.39	0.02	1.47 (1.43–1.52)	<0.001
Age group				
25–29			Ref.	0.011
30–39	0.25	0.03	1.29 (1.21–1.36)	<0.001
40–49	0.36	0.03	1.43 (1.34–1.52)	<0.001
50–59	0.53	0.03	1.69 (1.59–1.80)	<0.001
≥60	0.42	0.03	1.52 (1.42–1.62)	<0.001
Residence				
Rural			Ref.	
City	0.05	0.02	1.06 (1.02–1.09)	0.003
Education				
Middle school	0.71	0.06	2.03 (1.82–2.26)	<0.001
High school	0.64	0.05	1.89 (1.70–2.10)	<0.001
Undergraduate	0.52	0.05	1.69 (1.52–1.87)	<0.001
Graduate			Ref.	0.15

**Table 4 ijerph-19-02096-t004:** Comparison of blood parameters between participants with different BMI.

Variables	Lean/Normal Weight MAFLD (%)	Overweight MAFLD (%)	Obese MAFLD (%)	Non-MAFLD (%)	*p*
N	724 (0.98)	2012 (2.74)	21,096 (28.68)	49,734 (67.60)	
Age group					
25–29	115 (15.88)	205 (10.19)	2029 (9.62)	7467 (15.01)	<0.001
30–39	148 (20.44)	404 (20.08)	4306 (20.41)	11,731 (23.59)	
40–49	111 (15.33)	372 (18.49)	3930 (18.63)	9132 (18.36)	
50–59	148 (20.44)	483 (24.01)	5372 (25.46)	10,068 (20.24)	
≥60	202 (27.90)	548 (27.24)	5459 (25.88)	11,336 (22.79)	
Gender					
Male	272 (37.57)	1019 (50.65)	11,161 (52.91)	21,388 (43.00)	<0.001
Female	452 (62.43)	993 (49.35)	9935 (47.09)	28,346 (56.99)	
Hypertension (%)					
No	515 (71.13)	1352 (67.20)	11,570 (54.85)	36,358 (73.10)	<0.001
Yes	209 (28.87)	660 (32.80)	9526 (45.15)	13,376 (26.90)	
Diabetes DM (%)					
No	635 (87.71)	1649 (81.96)	16,844 (79.84)	45,387 (91.26)	<0.001
Yes	89 (12.29)	363 (18.04)	4252 (20.16)	4347 (8.74)	
Waist circumference (cm)	80.17 ± 7.53 ^a^	85.36 ± 6.60 ^ab^	95.604 ± 8.88 ^abc^	82.17 ± 9.73 ^abc^	<0.001
Hip circumference (cm)	93.90 ± 5.41 ^a^	96.94 ± 5.38 ^ab^	104.78 ± 7.17 ^abc^	95.96 ± 6.97 ^abc^	<0.001
ALT (IU/L)	23.78 ± 44.53 ^a^	25.43 ± 17.39 ^ab^	30.22 ± 25.67 ^abc^	18.87 ± 14.80 ^abc^	<0.001
AST (IU/L)	23.96 ± 30.30 ^a^	23.33 ± 15.79 ^b^	24.72 ± 15.63 ^bc^	20.94 ± 10.86 ^abc^	<0.001
TP (g/L)	78.30 ± 4.93 ^a^	78.41 ± 4.89 ^bc^	78.15 ± 4.91 ^bc^	77.53 ± 4.88 ^abc^	<0.001
ALB (g/L)	48.32 ± 3.50 ^a^	48.76 ± 3.467 ^ab^	48.60 ± 3.25 ^abc^	48.75 ± 3.11 ^ac^	<0.001
TBIL (umol/L)	12.25 ± 5.42	12.56 ± 5.33 ^b^	12.17 ± 5.20 ^bc^	12.43 ± 5.14 ^c^	<0.001
DBIL (umol/L)	3.47 ± 2.14 ^a^	3.56 ± 1.84 ^b^	3.53 ± 1.81 ^c^	3.66 ± 1.77 ^abc^	<0.001
IBIL (umol/L)	8.78 ± 3.83	9.01 ± 3.83 ^b^	8.64 ± 3.82 ^bc^	8.77 ± 3.75 ^bc^	<0.001
ALP (IU/L)	86.94 ± 31.32 ^a^	89.57 ± 25.05 ^ab^	91.45 ± 26.50 ^abc^	83.65 ± 26.20 ^abc^	<0.001
GGT (IU/L)	38.89 ± 86.56 ^a^	37.60 ± 41.69 ^b^	41.77 ± 39.99 ^abc^	26.95 ± 33.73 ^abc^	<0.001
TBA (umol/L)	4.29 ± 5.55 ^a^	4.03 ± 4.94 ^b^	4.08 ± 4.58 ^c^	3.70 ± 4.10 ^abc^	<0.001
CHE (IU/L)	8341.40 ± 1608.05 ^a^	8814.04 ± 1415.35 ^ab^	9064.56 ± 1465.08 ^abc^	7993.57 ± 1474.81 ^abc^	<0.001
PA (g/L)	0.29 ± 0.06 ^a^	0.30 ± 0.05 ^ab^	0.30 ± 0.05 ^abc^	0.28 ± 0.05 ^abc^	<0.001
TG (umol/L)	1.79 ± 1.75 ^a^	2.06 ± 1.98 ^ab^	2.29 ± 2.26 ^abc^	1.37 ± 1.26 ^abc^	<0.001
FPG (mmol/L))	6.04 ± 1.97 ^a^	6.24 ± 2.01 ^ab^	6.43 ± 2.13 ^abc^	5.75 ± 1.60 ^abc^	<0.001
CHOL (mmo/L)	5.06 ± 1.08 ^a^	5.20 ± 1.03 ^ab^	5.23 ± 1.06 ^ac^	4.99 ± 0.99 ^bc^	<0.001
HDL-C (mmo/L)	1.41 ± 0.37 ^a^	1.31 ± 0.31 ^ab^	1.25 ± 0.29 ^abc^	1.43 ± 0.34 ^bc^	<0.001
LDL-C (mmo/L)	2.92 ± 0.86 ^a^	3.10 ± 0.84 ^ab^	3.15 ± 0.85 ^c^	2.87 ± 0.82 ^bc^	<0.001
HbA1c (%)	5.59 ± 1.14 ^a^	5.72 ± 1.18 ^ab^	5.83 ± 1.13 ^abc^	5.44 ± 0.88 ^abc^	<0.001
RBC× (10^12^/L)	4.65 ± 0.46 ^a^	4.78 ± 0.46 ^ab^	4.83 ± 0.46 ^abc^	4.66 ± 0.45 ^bc^	<0.001
HGB (g/L)	132.74 ± 15.31 ^a^	137.34 ± 14.94 ^ab^	139.50 ± 14.84 ^abc^	134.34 ± 15.25 ^abc^	<0.001
HCT (%)	0.39 ± 0.04 ^a^	0.404 ± 0.035 ^ab^	0.41 ± 0.04 ^abc^	0.39 ± 0.04 ^abc^	<0.001
MCV (fl)	84.49 ± 4.92	84.36 ± 4.58 ^b^	84.49 ± 4.21 ^c^	84.82 ± 4.73 ^bc^	<0.001
MCH (pg)	28.60 ± 2.25 ^a^	28.81 ± 2.16	28.93 ± 4.80 ^a^	28.90 ± 2.78 ^a^	0.043
MCHC (g/L)	338.14 ± 13.08 ^a^	341.20 ± 12.97 ^a^	342.14 ± 56.89 ^ac^	340.48 ± 22.57 ^c^	<0.001
RDW (%)	0.11 ± 0.02	0.11 ± 0.01 ^b^	0.11 ± 0.01 ^bc^	0.11 ± 0.01 ^bc^	0.001
PLT (×10^9^/L)	199.56 ± 56.19 ^a^	210.80 ± 55.96 ^ab^	221.50 ± 58.65 ^abc^	217.28 ± 53.59 ^abc^	<0.001
PDW (%)	0.20 ± 0.01 ^a^	0.20 ± 0.01	0.20 ± 0.01 ^ac^	0.20 ± 0.01 ^c^	0.002
MPV (fl)	6.32 ± 1.07 ^a^	6.39 ± 1.12	6.40 ± 1.11	6.41 ± 1.13 ^a^	0.21
PCT (%)	0.12 ± 0.03 ^a^	0.13 ± 0.03 ^ab^	0.14 ± 0.07 ^abc^	0.14 ± 0.04 ^abc^	<0.01
WBC (×10^9^/L)	6.20 ± 1.74 ^a^	6.44 ± 1.63 ^ab^	6.80 ± 1.70 ^abc^	6.15 ± 1.60 ^bc^	<0.001
LYM (×10^9^/L)	2.16 ± 0.77 ^a^	2.17 ± 0.61 ^b^	2.31 ± 0.68 ^abc^	2.08 ± 0.64 ^abc^	<0.001
MONO (×10^9^/L)	0.38 ± 0.15 ^a^	0.40 ± 0.15 ^ab^	0.42 ± 0.16 ^abc^	0.39 ± 0.16 ^bc^	<0.001
NEU (×10^9^/L)	3.48 ± 1.33 ^a^	3.67 ± 1.26 ^ab^	3.85 ± 1.27 ^abc^	3.50 ± 1.23 ^bc^	<0.001
EOS (×10^9^/L)	0.13 ± 0.12 ^a^	0.15 ± 0.15 ^ab^	0.16 ± 0.16 ^abc^	0.14 ± 0.14 ^bc^	<0.001
BASO (×10^9^/L)	0.05 ± 0.04 ^a^	0.06 ± 0.04 ^b^	0.06 ± 0.07 ^abc^	0.06 ± 0.04 ^c^	<0.001
Alcohol consumption (g)	6.53 ± 34.28	7.82 ± 25.20 ^b^	8.14 ± 24.40 ^c^	5.93 ± 20.69 ^bc^	<0.001
LSM (Kpa)	6.80 ± 5.27 ^a^	6.74 ± 3.54 ^b^	7.50 ± 3.03 ^abc^	6.10 ± 2.55 ^abc^	<0.001
UAP (dB/m)	263.46 ± 18.78 ^a^	265.67 ± 19.14 ^ab^	276.32 ± 25.90 ^abc^	208.61 ± 20.81 ^abc^	<0.001

The same superscript letters in two groups represent *p* value < 0.05 when the two groups are compared. ^a^ Compared with lean/normal weight MAFLD groups, *p* value < 0.05; ^b^ Compared with overweight MAFLD groups, *p* value < 0.05; ^c^ Compared with obese MAFLD groups, *p* value < 0.05.

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
