# Peer review of "Prevalence and Risk Factors of Metabolic-Associated Fatty Liver Disease among 73,566 Individuals in Beijing, China"

_ijerph, 2022, doi:10.3390/ijerph19042096_

Round 1

Reviewer 1 Report

This was a large community study conducted in Beijing China of over 70,000 adults over 24 years of age to evaluate prevalence and risk factors of Metabolic associated fatty liver disease (MAFLD).  MAFLD was diagnosed using transient elastography plus the presence of any one of the following three conditions (overweight/obesity, diabetes, evidence of metabolic dysregulation) based on new 2020 consensus definition.  The data collection and blood sample analysis were comprehensive with the exception of CRP and insulin - as pointed out by the authors in the discussion of limitations.  

Specific comments:

  1.   Line 45 to 50 reads like marketing copy and should be deleted.  "As a new generation of TE, FibroTouch can provide both fat attenuation parameter (FAP) and liver fibrosis (LSM), which has demonstrated good accuracy in quantifying the levels of liver steatosis and fibrosis in MAFLD patients. FibroTouch has a low failure rate with moderate to high diagnostic performance for discriminating the steatosis degree and fibrosis stage and is suitable for clinical evaluation and monitoring of patients with MAFLD."
  2. Nearly 50% of the references are prior to 2015.  Please check if there are more appropriate/recent or if comparisons are appropriate given changes in definitions and diagnostic tests used.

Author Response

Point 1: Line 45 to 50 reads like marketing copy and should be deleted. "As a new generation of TE, FibroTouch can provide both fat attenuation parameter (FAP) and liver fibrosis (LSM), which has demonstrated good accuracy in quantifying the levels of liver steatosis and fibrosis in MAFLD patients. FibroTouch has a low failure rate with moderate to high diagnostic performance for discriminating the steatosis degree and fibrosis stage and is suitable for clinical evaluation and monitoring of patients with MAFLD."

Response 1: Thank you for the reviewers’ warm comments concerning our manuscript, and we have deleted the sentence in line 45 to 50 as suggested.

Point 2: Nearly 50% of the references are prior to 2015. Please check if there are more appropriate/recent or if comparisons are appropriate given changes in definitions and diagnostic tests used.

Response 2: Thank you very much for your kind reminding. We have checked the latest literatures and updated some of our references accordingly as suggested. Most of the literatures related to MAFLD published so far were based on the old definition(NAFLD) and few surveys used the new definition. Referring to other published literatures on MAFLD, we also made comparisons given changes in definitions and diagnostic tests in this study.

Special thanks for your good comments.

Reviewer 2 Report

MAFLD, a chronic liver disease, is frequently observed in elderly people and in patients with obesity, type 2 diabetes mellitus, hypertension, and metabolic syndrome. Thus, understanding the risk factors of MAFLD is help to optimize choices of treatment, follow-up monitoring, and eventually identify modifiable risk factors (e.g., diet, physical activity, etc.). In this manuscript, the authors represented an interesting results that NAFLD associated with male, older age, lower education and urban 23 residence. In addition, participants with lean and normal-weights was under risk of MAFLD. This is well constructed cross-sectional study; however, I think, there are some concerns that need to be addressed. Specific comments are as follows:

Abstract

  • Line 9, 11, 13, 21: Delete headings

Introduction

Please divide the paragraphs in the introduction appropriately. I recommend dividing the paragraph as follows:

  • in line 33: MAFLD. / Over the past two ~
  • in line 41: in the last decades. / Liver biopsy ~
  • in line 50: patients with MAFLD. / It is very~

Methods

  • Line 59: The stratified sampling method is the most practically used extraction method in the sampling method. Please explain in a little more detail about the stratification method of the sampling process and the number of samples you have taken.
  • Line 73~74: Describe the full words instead of shorten form.
  • Line 77: Please add the country of your manufacturing company.
  • Line 102: This study aimed to estimate the prevalence and risk factors of MAFLD in adult population of Beijing. In this case, is there any reason to subdivide the MAFLD group by BMI? Please additionally explain the importance or reason in Discussion section.
  • Line 110: If you are diagnosed with high blood pressure and take medications, your blood pressure may be controlled. Is it not included in the case of taking high blood pressure medications?
  • Line 118: If the sample is stratified as described in line59, statistical analysis should be performed taking into account stratification, cluster, and weights. Please describe whether weights were considered in statistical analysis.

Results

  • Line 131~134: I recommend to move in the section ‘2.1 study population and data collection’.
  • Line 138, Table 1: Is it needed to describe the value of χ2?
  • Line 155, Table 2: Is it needed to describe the value of T?
  • Line 186, Table 4: Is it needed to describe the value of χ2 or F?

Discussion

  • Line 195~199: I recommend that the information on the world and that on China are described separately.
  • Line 202~204: I recommend that you describe about the interactions between estrogen and MAFLD in the section.

Supplementary materials: If it doesn't apply to this study, please delete it.

Institutional Review Board Statement: I think that this study was approved by the institutional review board and provided written informed content from all participants. Please describe the information.

Informed consent statement: Same as ‘Institutional review board statement’

Data Availability Statement: If it doesn't apply to this study, please delete it.

Author Response

Response to Reviewer 2 comments

Abstract

Point 1: Line 9, 11, 13, 21: Delete headings

Response 1: Thank you for the reviewers’ comments concerning our manuscript, we have deleted headings mentioned in line Line 9, 11, 13, 21.

Introduction

Point 2: Please divide the paragraphs in the introduction appropriately. I recommend dividing the paragraph as follows…

Response 2: Thank you for the comments on the paper. We have divided the paragraphs in the introduction as suggested.

Methods

Point 3: Line 59: The stratified sampling method is the most practically used extraction method in the sampling method. Please explain in a little more detail about the stratification method of the sampling process and the number of samples you have taken.

Response 3: Thank the reviewer for the comments and we revised the manuscript accordingly. In the revised paper, we have added the detailed explanation to the sampling method and the number of samples.

Point 4: Line 73~74: Describe the full words instead of shorten form.

Response 4: Thank you for the warm comments on the paper and we have revised the manuscript by using full words instead of shorten form as suggested.

Point 5: Line 77: Please add the country of your manufacturing company.

Response 5: Thank you for the comments and we have added the country information as suggested.

Point 6: Line 102: This study aimed to estimate the prevalence and risk factors of MAFLD in adult population of Beijing. In this case, is there any reason to subdivide the MAFLD group by BMI? Please additionally explain the importance or reason in Discussion section.

Response 6: Yes, the reasons for subdividing the MAFLD group by BMI were as follows: 1) It is essential to distinguish normal weight and lean from overweight-obese MAFLD, so that specific treatment can be provided to halt or prevent the development of MAFLD. 2) The lack of knowledge of the prevalence and characteristics of lean/normal weight MAFLD in the Chinese population prompted us to define the prevalence and characterize the clinical biochemical, blood cell and metabolic features of lean/normal weight MAFLD. 

Thank you for the comments and we have added explaination of the reason to subdivide the MAFLD group by BMI in the Discussion section as suggested.

Line 110: If you are diagnosed with high blood pressure and take medications, your blood pressure may be controlled. Is it not included in the case of taking high blood pressure medications?

Response 7: Yes. When diagnosing hypertension, those who gave a “yes” answer to the question of “Have you taken any anti-hypertensive drugs in the last six months?” were excluded from analysis. We have supplemented the descriptions in the “Diagnostic criteria” section.

Point 8: Line 118: If the sample is stratified as described in line 59, statistical analysis should be performed taking into account stratification, cluster, and weights. Please describe whether weights were considered in statistical analysis.

Response 8: Thank the reviewer for the comments. In this study, the probability proportional to size (PPS) method was used to sampling township and residents' committees/villages. PPS sampling was characterized by that the probability of selecting a unit is proportional to its size, which guaranteed the representativeness of the sample. Referring to the analytical methods of other similar studies published, we did statistical analysis at the individual level in this manuscript.

Results

Point 9:Line 131~134: I recommend to move in the section ‘2.1 study population and data collection’.

Response 9: Thank the reviewer for the recommendations and we have revised the ‘2.1 study population and data collection’ part by adding descriptions about the number of samples.

Point 10: Line 138, Table 1: Is it needed to describe the value of χ2?

Point 11: Line 155, Table 2: Is it needed to describe the value of T?

Point 12: Line 186, Table 4: Is it needed to describe the value of χ2 or F?

Response 10-12: Thank you for your comments and we have deleted the columns where χ2 or T or F existed in.

Discussion

Point 13: Line 195~199: I recommend that the information on the world and that on China are described separately.

Response 13: Thank you very much for your warm reminding. The prevalence of MAFLD in the world was presented in the Introduction section, and we mainly intended to compared the prevalence rate of MAFLD in Beijing and rates reported in other places of China in the Discussion part, so only information on in China were presented here.

Point 14: Line 202~204: I recommend that you describe about the interactions between estrogen and MAFLD in the section.

Response 14: Thank the reviewer for the comments and we have supplemented descriptions about the interactions between estrogen and MAFLD in the Discussion section as suggested.

Point 15: Supplementary materials: If it doesn't apply to this study, please delete it.

Response 15: Thank the reviewer for the warm comments and we have deleted it.

Point 16: Institutional Review Board Statement: I think that this study was approved by the institutional review board and provided written informed content from all participants. Please describe the information.

Point 17: Informed consent statement: Same as ‘Institutional review board statement’

Response 16-17: Thank the reviewer for the kind comments and we have added the information of Institutional Review Board Statement and Informed consent statement as suggested.

Point 18: Data Availability Statement: If it doesn't apply to this study, please delete it.

Response 18: Thank the reviewer for the comments and we have deleted it.

Special thanks for all your comments.

Reviewer 3 Report

The present article title “Prevalence and Risk Factors of Metabolic Associated Fatty Liver Disease among 73566 individuals in Beijing, China” by Qianli Yuan, Huai Wang, Pei Gao, Weixin Chen, Min Lv, Shuang Bai, Jiang Wu, showed the prevalence of MAFLD, and the mainly risk factors associated with this pathology.

Minor comments

Page 2, line 57

It is important to mention that if any ethics committee approved the study.

 Page 2 lines 69-75

The authors could mention under what conditions the patient presented for blood sampling. For example, if the patients were fasting of 10 h or in postprandial condition.

The autors should defined the following term ALT, AST, FPG, GGT, ALB, TG, TC, CHOL, 73 HDL-C, LDL-C, HbAlc, AFP, WBC, viral markers, etc. All

Page 3, line 110

The term diabetes mellitus was defined previously as DM. Also, the authors may revise the term throughout the text to unify it.

Page 3, line 119

The phrase meanstandard deviation (xs) may be modified by mean  standard deviation

Page 3, line 131

 The autor may modified the following sensense “74,998 participants were involved in this study, …” by “In this study were involved 74,998 participants, …. “

Page 3-4

In the results of table 1, a suggestion for the author is that they may consider performing the Ch2 test for trend, since in some of the analyses more than 2 variables are considered and the Chi-square test only considers 2x 2 analyses.

Author Response

Response to Reviewer 3 comments

Point 1: It is important to mention that if any ethics committee approved the study.

Response 1: Thank you for the reviewers’ comments. This study was approved by ethics review committee of Beijing Centers for Disease Control and Prevention, and the approval number was No: 2017(7th). The informed consent was obtained from all subjects involved in the study. The above mentioned information have been added to the ”Institutional Review Board Statement” and ”Informed Consent Statement” respectively as suggested.

Point 2: The authors could mention under what conditions the patient presented for blood sampling. For example, if the patients were fasting of 10 h or in postprandial condition.

Response 2: Thank you very much for your comments. We have supplemented descriptions in the “Blood collection and testing” section as suggested.

Point 3:The autors should defined the following term ALT, AST, FPG, GGT, ALB, TG, TC, CHOL, 73 HDL-C, LDL-C, HbAlc, AFP, WBC, viral markers, etc. All

Response 3: Thank you for the warm comments on the paper and we have revised the manuscript by using full words instead of shorten form.

Point 4: The term diabetes mellitus was defined previously as DM. Also, the authors may revise the term throughout the text to unify it.

Response 4: Thank you so much for your reminding and we have corrected the mistakes.

Point 5: The autor may modified the following sensense “74,998 participants were involved in this study, …” by “In this study were involved 74,998 participants, …. “

Response 5: Thank you for reminding us the improper descriptions on the study. We have the improper sentences revised accordingly.

Point 6: In the results of table 1, a suggestion for the author is that they may consider performing the Ch2 test for trend, since in some of the analyses more than 2 variables are considered and the Chi-square test only considers 2x 2 analyses.

Response 6: Thank you very much for your warm comments and we have updated the results in table 1 by using Ch2 test for trend as suggested, and updated descriptions about analysis method in“2.5. statistical analysis” section. Compared results from Chi-square test, value of χ2 from Ch2 test for trend method actually decreased, while all P value still less than 0.001. In this revised version of manuscript, value of χ2, T or F were recommended not be presented by one of the reviewer, so the original columns where χ2 or T or F existed in were deleted.

Special thanks for all your good comments.